# Euclidean Equivariant Models for Generative Graphical Inverse Kinematics

Oliver Limoyo,[1,†] Filip Marić,[1,2,†] Matthew Giamou,[1] Petra Alexson,[1] Ivan Petrović,[2] and Jonathan Kelly[1]

[†]Denotes equal contribution.
[1]Institute for Aerospace Studies, University of Toronto,
[2]Laboratory for Autonomous Systems and Mobile Robotics, University of Zagreb

*Abstract*—Quickly and reliably finding accurate inverse kine-matics (IK) solutions remains a challenging problem for robotic manipulation. Existing numerical solvers typically produce a single solution only and rely on local search techniques to minimize a highly nonconvex objective function. Recently, learning-based approaches that approximate the entire feasible set of solutions have shown promise as a means to generate multiple fast and accurate IK results in parallel. However, existing learning-based techniques have a significant drawback: each robot of interest requires a specialized model that must be trained from scratch. To address this shortcoming, we investigate a novel distance-geometric robot representation coupled with a graph structure that allows us to leverage the flexibility of graph neural networks (GNNs). We use this approach to train a generative graphical inverse kinematics solver (GGIK) that is able to produce a large number of diverse solutions in parallel while also generalizing well—a single learned model can be used to produce IK solutions for a variety of different robots. The graphical formulation elegantly exposes the symmetry and Euclidean equivariance of the IK problem that stems from the spatial nature of robot manipulators. We exploit this symmetry by encoding it into the architecture of our learned model, yielding a flexible solver that is able to produce sets of IK solutions for multiple robots.

## I. Introduction

Robotic manipulation tasks are naturally defined in terms of end-effector poses (for, e.g., bin-picking or path following). However, the configuration of a manipulator is typically specified in terms of joint angles, and determining the joint configuration(s) that correspond to a given end-effector pose requires solving the *inverse kinematics* (IK) problem. For redundant manipulators (i.e., those with more than six degrees of freedom, or DOF), target poses may be reachable by an infinite set of feasible configurations. While redundancy allows high-level algorithms such as motion planners to choose configurations that best fit the overall task, it makes solving IK substantially more involved.

Since the full set of IK solutions cannot, in general, be derived analytically for redundant manipulators, individual configurations reaching a target pose are found by locally searching the configuration space using numerical optimization methods and geometric heuristics. These limitations have motivated the use of learned models that approximate the entire feasible set of solutions. In terms of success rate, learned models that output individual solutions are able to compete with the best numerical IK solvers when high accuracy is not required [19]. Data-driven methods are also useful for integrating abstract criteria such as "human-like" poses or motions [2].

Generative approaches [8, 15] have demonstrated the ability to rapidly produce a large number of approximate IK solutions and even model the entire feasible set for specific robots [1]. Unfortunately, these learned models, parameterized by deep neural networks (DNNs), require specific configuration and end-effector input-output vector pairs for training (by design). In turn, it is not possible to generalize learned solutions to robots that vary in link geometry or DOF. Ultimately, this drawback limits the utility of learning for IK over well-established numerical methods that are easier to implement and generalize [3].

In this paper, we describe a novel generative inverse kinematics solver and explain its capacity to simultaneously represent general (i.e., not tied to a single robot manipulator model or geometry) IK mappings and to produce approximations of entire feasible sets of solutions. In contrast to existing DNN-based approaches [1, 8, 11, 15, 19], we explore a new path towards learning generalized IK by adopting a *graphical* model of robot kinematics [13, 14]. This graph-based description allows us to make use of graph neural networks (GNNs) to capture varying robot geometries and DOF within a single model. Furthermore, crucial to the success of our method, the graphical formulation exposes the symmetry and Euclidean equivariance of the IK problem that stems from the spatial nature of robot manipulators. We exploit this symmetry by encoding it into the architecture of our learned model, which we call GGIK (for *generative graphical inverse kinematics*), to produce accurate IK solutions.

## II. Graph Representation for Inverse Kinematics

The mapping $IK : \mathcal{T} \to \mathcal{C}$ from task space $\mathcal{T}$ to configuration space $\mathcal{C}$ defines the *inverse kinematics* of the robot, connecting a target pose $\mathbf{T} \in \mathrm{SE}(3)$ to one or more feasible configurations $\boldsymbol{\theta} \in \mathcal{C}$. In this paper, we consider the associated problem of determining this mapping for manipulators with $n > 6$ DOF (also known as *redundant* manipulators), where each end-effector pose corresponds to a set of configurations

$$IK(\mathbf{T}) = \{\boldsymbol{\theta} \in \mathcal{C} \mid FK(\boldsymbol{\theta}) = \mathbf{T}\} \qquad (1)$$

that we refer to as the full set of IK solutions.

We eschew the common angle-based representation of the configuration space in favour of a *distance-geometric* model of robotic manipulators comprised of revolute joints [14]. This allows us to represent configurations $\boldsymbol{\theta}$ as complete graphs

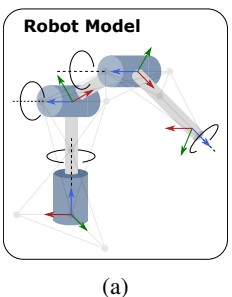

**Robot Model**

(a)

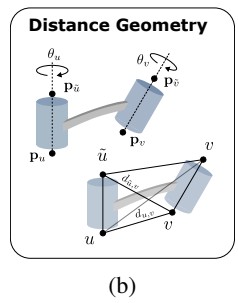

**Distance Geometry**

(b)

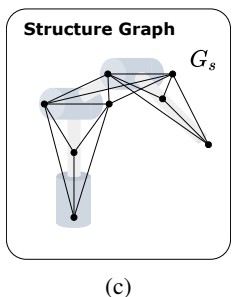

**Structure Graph**

$G_s$

(c)

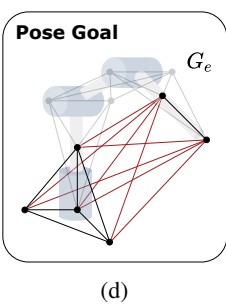

**Pose Goal**

$G_e$

(d)

Fig. 1: The process of defining an IK problem as an incomplete or *partial* graph $\widetilde{G}$ of inter-point distances. (a) Conventional forward kinematics model parameterized by joint angles and joint rotation axes. (b) The point placement procedure for the distance based description, first introduced in [13]. Note that the four distances between points associated with pairs of consecutive joints remain constant regardless of the of configuration. (c) A structure graph of the robot based on inter-point distances. (d) Addition of distances in red describing the robot end-effector pose using auxiliary points to define the base coordinate system, completing the graphical IK problem description. All configurations of the robot reaching this end-effector pose will result in a partial graph of distances shown in (c) and (d).

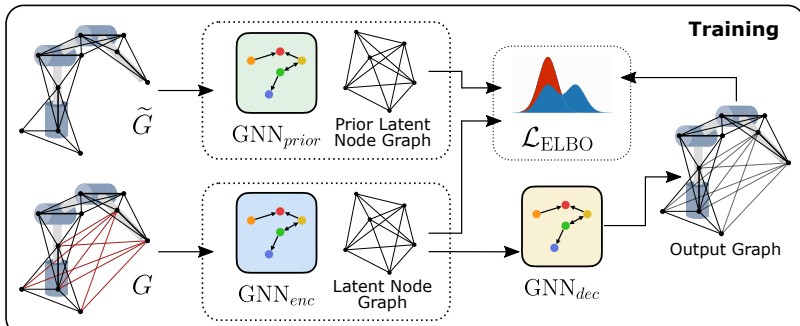

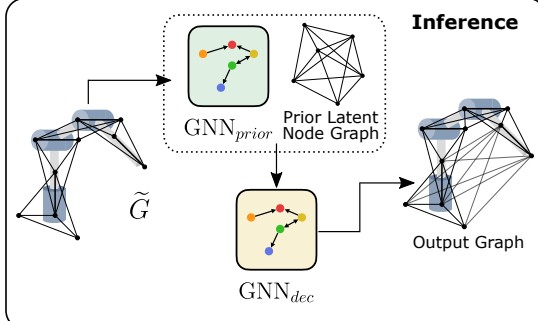

Fig. 2: Our GGIK solver is based on the CVAE framework. $\text{GNN}_{enc}$ encodes a complete manipulator graph into a latent graph representation and $\text{GNN}_{dec}$ "reconstructs" it. The prior network, $\text{GNN}_{prior}$, encodes the partial graph into a latent embedding that is near the embedding of the full graph. At inference time, we decode the latent embedding of a partial graph into a complete graph to generate a solution.

$G = (V, E)$. The edges $E$ are weighted by distances $d$ between a collection of $N$ points $\mathbf{p} = \{\mathbf{p}_i\}_{i=1}^N \in \mathbb{R}^{N \times D}$ indexed by vertices $V$, where $D \in \{2, 3\}$ is the workspace dimension. The coordinates of points corresponding to these distances are recovered by solving the distance geometry problem (DGP):

**Distance Geometry Problem** ([12]). *Given an integer $D > 0$, a set of vertices $V$, and a simple undirected graph $G = (V, E)$ whose edges $\{u, v\} \in E$ are assigned non-negative weights $\{u, v\} \mapsto d_{u,v} \in \mathbb{R}_+$, find a function $p : V \to \mathbb{R}^D$ such that the Euclidean distances between neighbouring vertices match their edges' weights (i.e., $\forall \{u, v\} \in E, \|p(u) - p(v)\| = d_{u,v}$).*

It was shown in [13] that any solution $\mathbf{p} \in DGP(G)$ may be mapped to a unique corresponding configuration $\boldsymbol{\theta}$.[1] Crucially, this allows us to a construct a partial graph $\widetilde{G} = (V, \widetilde{E})$, with $\widetilde{E} \subset E$ corresponding to distances determined by an end-effector pose $\mathbf{T}$ and the robot's structure (i.e., those common to all elements of $IK(\mathbf{T})$), where each $\mathbf{p} \in DGP(\widetilde{G})$ corresponds to a particular IK solution $\boldsymbol{\theta} \in IK(\mathbf{T})$. The generic procedure for constructing $\widetilde{G}$ is demonstrated for a simple manipulator in Fig. 1. A more detailed overview of the distance-geometric graph representation and graph construction is available in [13].

[1]Up to any Euclidean transformation of $\mathbf{p}$, since distances are invariant to such a transformation.

For a complete graph $G$, we define the GNN node features as a combination of point positions $\mathbf{p} = \{\mathbf{p}_i\}_{i=1}^N \in \mathbb{R}^{N \times D}$ and general features $\mathbf{h} = \{\mathbf{h}_i\}_{i=1}^N$, where each $\mathbf{h}_i$ is a feature vector containing extra information about the node. We use a three-dimensional one-hot-encoding, $\mathbf{h}_i \in \{0, 1\}^3$ and $\sum_{j=1}^3 h_{i,j} = 1$, that indicates whether the node defines the base coordinate system, a general joint or link, or the end-effector. Similarly, we define the $M$ known point positions of the partial graph $\widetilde{G}$ as $\tilde{\mathbf{p}} = \{\tilde{\mathbf{p}}_i\}_{i=1}^M \in \mathbb{R}^{M \times D}$ and set the remaining unknown $N - M$ node positions to zero. The partial graph shares the same general features $\mathbf{h}$ as the complete graph. In both cases, the edge features are simply the corresponding inter-point distances between known node point positions or initialized to zero if unknown.

## III. GENERATIVE GRAPHICAL INVERSE KINEMATICS

At its core, GGIK is a conditional variational autoencoder (CVAE) model [17] that parameterizes the conditional distribution $p(G \mid \widetilde{G})$ using GNNs. By introducing an unobserved stochastic latent variable $\mathbf{z}$, our generative model is defined as

$$p_\gamma(G \mid \widetilde{G}) = \int p_\gamma(G \mid \widetilde{G}, \mathbf{z}) \, p_\gamma(\mathbf{z} \mid \widetilde{G}) \, d\mathbf{z}, \qquad (2)$$

where $p_\gamma(G \mid \widetilde{G}, \mathbf{z})$ is the likelihood of the full graph, $p_\gamma(\mathbf{z} \mid \widetilde{G})$ is the prior, and $\gamma$ are the learnable generative parameters. The

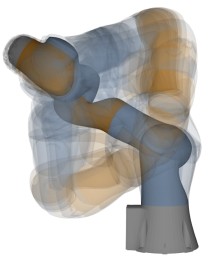 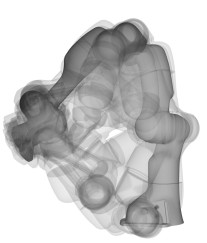 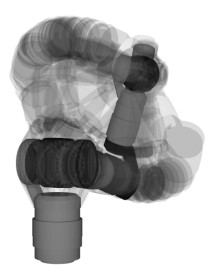 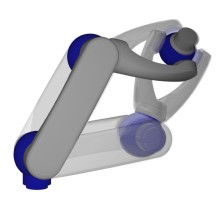 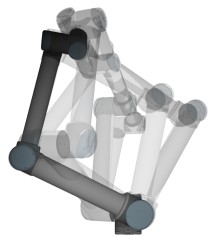

Fig. 3: Sampled conditional distributions from GGIK for various robotic manipulators. From left to right: KUKA IIWA, Franka Emika Panda, Schunk LWA4D, Schunk LWA4P, and Universal Robots UR10. Note that the end-effector poses are nearly identical in all cases, highlighting kinematic redundancy. Our model is able to capture the discrete solution set for the two non-redundant robots as well.

likelihood is given by

$$p_\gamma(G \,|\, \widetilde{G}, \mathbf{z}) = \prod_{i=1}^{N} p_\gamma(\mathbf{p}_i \,|\, \widetilde{G}, \mathbf{z}_i), \quad \text{with}$$
$$p_\gamma(\mathbf{p}_i \,|\, \widetilde{G}, \mathbf{z}_i) = \mathcal{N}(\mathbf{p}_i \,|\, \boldsymbol{\mu}_i, \mathbf{I}), \tag{3}$$

where $\mathbf{p} = \{\mathbf{p}_i\}_{i=i}^{N}$ are the positions of all $N$ nodes, $\mathbf{z} = \{\mathbf{z}_i\}_{i=i}^{N}$ are the latent embeddings of each node, and $\boldsymbol{\mu} = \{\boldsymbol{\mu}_i\}_{i=i}^{N}$ are the predicted means of the distribution of node positions. We parametrize the likelihood distribution with a GNN decoder, that is, $\boldsymbol{\mu}$ is the output of $\text{GNN}_{dec}(\widetilde{G}, \mathbf{z})$. In practice, for the input of $\text{GNN}_{dec}(\cdot)$, we concatenate each latent node with the respective position node features $\tilde{\mathbf{p}}$ of the original partial graph $\widetilde{G}$ when available and the general features $\mathbf{h}$. If unavailable, we concatenate the latent nodes with the initialized point positions set to zero. The prior distribution is given by

$$p_\gamma(\mathbf{z} \,|\, \widetilde{G}) = \prod_{i=1}^{N} p_\gamma(\mathbf{z}_i \,|\, \widetilde{G}), \quad \text{with}$$
$$p_\gamma(\mathbf{z}_i \,|\, \widetilde{G}) = \sum_{k=1}^{K} \pi_{k,i} \, \mathcal{N}\left(\mathbf{z}_i \,|\, \boldsymbol{\mu}_{k,i}, \text{diag}(\boldsymbol{\sigma}_{k,i}^2)\right). \tag{4}$$

Here, we parameterize the prior as a Gaussian mixture model with $K$ components. Each Gaussian is in turn parameterized by a mean $\boldsymbol{\mu}_k = \{\boldsymbol{\mu}_{k,i}\}_{i=1}^{N}$, diagonal covariance $\boldsymbol{\sigma}_k = \{\boldsymbol{\sigma}_{k,i}\}_{i=1}^{N}$, and a mixing coefficient $\boldsymbol{\pi}_k = \{\pi_{k,i}\}_{i=1}^{N}$, where $\sum_{k=1}^{K} \pi_{k,i} = 1$, $i = 1, ..., N$. We chose a mixture model to have an expressive prior capable of capturing the latent distribution of multiple solutions. We parameterize the prior distribution with a multi-headed GNN encoder $\text{GNN}_{prior}(\widetilde{G})$

---

**Algorithm 1: GGIK**

**Parameters:** $\widetilde{G}, \mathbf{T}_{goal}, K, L$

**Result:** Solution configurations with the lowest pose error $\boldsymbol{\theta}^* \in \mathbb{R}^{K \times n_{joints}}$.

$\mathbf{z}_L \sim p_\gamma(\mathbf{z} \,|\, \widetilde{G})$      ▷ Sample $L$ latents $\mathbf{z}$ from $\text{GNN}_{prior}$.

$\mathbf{p}_L \sim p_\gamma(\mathbf{p} \,|\, \widetilde{G}, \mathbf{z}_L)$      ▷ Get $L$ solutions via $\text{GNN}_{dec}$.

$\boldsymbol{\theta}_L \leftarrow \text{fromPoints}(\mathbf{p}_L)$      ▷ Recover $L$ configurations.

$\boldsymbol{\theta}^* \leftarrow \text{selectSolution}(\mathbf{T}_{goal}, \boldsymbol{\theta}_L, K)$      ▷ Choose best $K$.

---

that outputs parameters $\{\boldsymbol{\mu}_k, \boldsymbol{\sigma}_k, \boldsymbol{\pi}_k\}_{k=1}^{K}$.

The goal of learning is to maximize the marginal likelihood or evidence of the data as shown in Eq. 2. As is commonly done in the variational inference literature [9], we instead maximize a tractable evidence lower bound (ELBO):

$$\mathcal{L} = \mathbb{E}_{q_\phi(\mathbf{z} \,|\, G)}[\log p(G \,|\, \widetilde{G}, \mathbf{z})] - KL(q_\phi(\mathbf{z} \,|\, G) || p_\gamma(\mathbf{z} \,|\, \widetilde{G})), \tag{5}$$

where $KL(\cdot || \cdot)$ is the Kullback-Leibler (KL) divergence and the inference model $q_\phi(\mathbf{z} \,|\, G)$ with learnable parameters $\phi$ is

$$q_\phi(\mathbf{z} \,|\, G) = \prod_{i=1}^{N} q_\phi(\mathbf{z}_i \,|\, G), \quad \text{with}$$
$$q_\phi(\mathbf{z}_i \,|\, G) = \mathcal{N}(\mathbf{z}_i \,|\, \boldsymbol{\mu}_i, \text{diag}(\boldsymbol{\sigma}_i^2)). \tag{6}$$

As with the prior distribution, we parameterize the inference distribution with a multi-headed GNN encoder, $\text{GNN}_{enc}(G)$, that outputs parameters $\boldsymbol{\mu} = \{\boldsymbol{\mu}_i\}_{i=1}^{N}$ and $\boldsymbol{\sigma} = \{\boldsymbol{\sigma}_i\}_{i=1}^{N}$. We summarize the full sampling procedure in Algorithm 1 and we visualize samples of these IK solutions in Fig. 3. This procedure can be done quickly and in parallel on the GPU.

## IV. E($n$) EQUIVARIANCE AND SYMMETRY

We are interested in mapping partial graphs $\widetilde{G}$ into full graphs $G$. Once trained, our model maps partial point sets to full point sets $f : \mathbb{R}^{M \times D} \rightarrow \mathbb{R}^{N \times D}$, where $f$ is a combination of networks $\text{GNN}_{prior}$ and $\text{GNN}_{dec}$ applied sequentially. The point positions (i.e., $\mathbf{p}$ and $\tilde{\mathbf{p}}$) of each node in the distance geometry problem contain underlying geometric relationships that we would like to preserve with our choice of architecture. Most importantly, the point sets are *equivariant* to the Euclidean group E($n$) of rotations, translations, and reflections. Let $S : \mathbb{R}^{M \times D} \rightarrow \mathbb{R}^{M \times D}$ be a transformation consisting of some combination of rotations, translations and reflections on the initial partial point set $\tilde{\mathbf{p}}$. Then, there exists an equivalent transformation $T : \mathbb{R}^{N \times D} \rightarrow \mathbb{R}^{N \times D}$ on the complete point set $\mathbf{p}$ such that:

$$f(S(\tilde{\mathbf{p}})) = T(f(\tilde{\mathbf{p}})). \tag{7}$$

To leverage this structure or geometric prior in the data, we use E($n$)-equivariant graph neural networks (EGNNs) [16] for $\text{GNN}_{dec}$, $\text{GNN}_{enc}$, and $\text{GNN}_{prior}$. The EGNN layer splits up the node features into an equivariant coordinate or position-based part and a non-equivariant part. We treat the positions

| Robot | Err. Pos. [mm] | | | | | Err. Rot. [deg] | | | | |
|---|---|---|---|---|---|---|---|---|---|---|
| | mean | min | max | $Q_1$ | $Q_3$ | mean | min | max | $Q_1$ | $Q_3$ |
| KUKA | 5.3 | 1.7 | 9.7 | 3.8 | 6.6 | 0.4 | 0.1 | 0.6 | 0.3 | 0.5 |
| Lwa4d | 4.7 | 1.4 | 9.1 | 3.2 | 5.9 | 0.4 | 0.1 | 0.6 | 0.3 | 0.5 |
| Lwa4p | 5.7 | 2.2 | 10.2 | 4.1 | 7.1 | 0.4 | 0.1 | 0.7 | 0.3 | 0.6 |
| Panda | 12.3 | 3.2 | 25.5 | 7.9 | 15.9 | 1.0 | 0.2 | 1.8 | 0.7 | 1.3 |
| UR10 | 9.2 | 4.2 | 14.7 | 7.3 | 11.1 | 0.5 | 0.2 | 0.9 | 0.4 | 0.7 |
| UR10 with DT [19] | 35.0 | - | - | - | - | 16.0 | - | - | - | - |
| Panda with IKFlow [1] | 7.7 | - | - | - | - | 2.8 | - | - | - | - |
| Panda with IKNet [4] | 31.0 | - | - | 13.5 | 48.6 | - | - | - | - | - |

TABLE I: Performance of GGIK on 2,000 randomly generated IK problems for a single model trained on five different robotic manipulators. Taking 32 samples from the learned distribution, the error statistics are presented as the mean and mean minimum and maximum error per problem and the two quartiles of the distribution. Note that all solutions were produced by a *single* GGIK model. We include baseline results from various other models that were trained on a single robot type. Dashed results were unavailable.

| Model Name | Err. Pos. [mm] | | | | | Err. Rot. [deg] | | | | | Test ELBO |
|---|---|---|---|---|---|---|---|---|---|---|---|
| | mean | min | max | $Q_1$ | $Q_3$ | mean | min | max | $Q_1$ | $Q_3$ | |
| EGNN [16] | 4.6 | 1.5 | 8.5 | 3.3 | 5.8 | 0.4 | 0.1 | 0.6 | 0.3 | 0.4 | -0.05 |
| MPNN [6] | 143.2 | 62.9 | 273.7 | 113.1 | 169.1 | 17.7 | 5.3 | 13.6 | 21.6 | 34.1 | -8.3 |
| GAT [18] | - | - | - | - | - | - | - | - | - | - | -12.41 |
| GCN [10] | - | - | - | - | - | - | - | - | - | - | -12.42 |
| GRAPHsage [7] | - | - | - | - | - | - | - | - | - | - | -10.5 |

TABLE II: Comparison of different network architectures. EGNN outperforms existing architectures that are not equivariant in terms of overall accuracy and test ELBO. Dashed results are models with output point sets that were too far from a valid joint configuration and diverged during the configuration reconstruction procedure.

$\mathbf{p}$ and $\tilde{\mathbf{p}}$ as the equivariant portion and the general features $\mathbf{h}$ as non-equivariant. As an example, a single EGNN layer $l$ from $\text{GNN}_{enc}$ is then defined as:

$$\begin{aligned}
\mathbf{m}_{ij} &= \phi_e(\mathbf{h}_i^l, \mathbf{h}_j^l, \|\mathbf{p}_i^l - \mathbf{p}_j^l\|^2) \\
\mathbf{p}_i^{l+1} &= \mathbf{p}_i^l + C \sum_{j \neq i} (\mathbf{p}_i^l - \mathbf{p}_j^l)\phi_x(\mathbf{m}_{ij}) \\
\mathbf{m}_i &= \sum_{j \neq i} \mathbf{m}_{ij} \\
\mathbf{h}_i^{l+1} &= \phi_h(\mathbf{h}_i^l, \mathbf{m}_i),
\end{aligned} \quad (8)$$

where, $\mathbf{m} \in \mathbb{R}^{f_m}$ with a message embedding dimension of $f_m$, $\phi_x : \mathbb{R}^{f_m} \to \mathbb{R}^1$, $C = \frac{1}{N-1}$ divides the sum by the number of elements, and $\phi_e$ and $\phi_h$ are typical edge and node operations approximated by multilayer perceptrons (MLPs). For more details about the model and a proof of the equivariance property, we refer readers to [16].

## V. EXPERIMENTS

We evaluate GGIK's capability to learn accurate solutions and generalize within a class of manipulator structures, and investigate the importance of capturing the Euclidean equivariance of the graphical formulation of inverse kinematics.

### A. Accuracy and Generalization

In Table I, we evaluate the accuracy of GGIK for a variety of existing commercial manipulators featuring different structures and numbers of joints: the Kuka IIWA, Schunk LWA4D, Schunk LWA4P, Universal Robots UR10, and Franka Emika Panda. We trained a single instance of GGIK on a total of 2,560,000 IK problems uniformly distributed over all five manipulators. We compare GGIK to other learned IK baselines [1, 4, 19] that are trained specifically for each robot. GGIK achieves better or comparable accuracy to all baselines despite generalizing across multiple manipulator types.

### B. Ablation Study on the Equivariant Network Architecture

We conducted an ablation experiment to evaluate the importance of capturing the underlying $\text{E}(n)$ equivariance of the distance geometry problem (Problem II) in our learning architecture. We compare the use of the EGNN network [16] to four common and popular GNN layers that are not $\text{E}(n)$ equivariant: GRAPHsage [7], GAT [18], GCN [10] and MPNN [6]. We match the number of parameters for each GNN architecture as closely as possible and keep all other experimental parameters fixed. Out of the five different architectures that we compare, only the EGNN and MPNN output point sets that can be successfully mapped to valid joint configurations. The equivariant EGNN model outperforms all other models in terms of the ELBO value attained on a held-out test set.

## VI. CONCLUSION

GGIK is a step towards learned "general" IK, that is, a solver (or initializer) that can provide multiple diverse solutions and can be used with any manipulator in a way that complements or replaces numerical optimization. The graphical formulation of IK naturally leads to the use of a GNN for learning, since the GNN can accept problems for arbitrary robots with different kinematic structures and degrees of freedom. Our formulation also exposes the Euclidean equivariance of the problem, which we exploit by encoding it into the architecture of our learned model. While our architecture demonstrates a capacity for generalization and an ability to produce diverse solutions, GGIK outputs may require post-processing via local optimization for applications with low error tolerances. As future work, we would like to learn constrained distributions of robot configurations that account for obstacles in the task space and for self-collisions; obstacles can be easily incorporated in the distance-geometric formulation of IK [5, 13].

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
