# OpenReview forum: "Euclidean Equivariant Models for Generative Graphical Inverse Kinematics"
_roboticsfoundation.org/RSS/2023/Workshop/Symmetry — RSS 2023 Workshop Symmetry_

### Official Review · Reviewer_nYJ1 · 2023-06-16
**This work proposed an approximated solution to solve the IK problem of different robot arms with the graph-based CVAE.**

**Rating:** 7
**Confidence:** 5

**Review:**

This paper proposed an approximated solution to solve the IK problem with the graph-based CVAE.
1. It represented the geometry of the robot with a distance-geometric model and define a graph to represent the configurations.
2. Based on the graph representation, the IK problem is transferred as distance geometry problem on the end-effector graph, i.e., $\tilde{G}$.
3. The full set of joint angles is modeled as $p(G|\tilde{G})$ which is parameterized by $GNN_{prior}$, $GNN_{enc}$ and $GNN_{dec}$.
4. It shows the benefits of the ENGNN on this geometric representation.

--------------------------------------------
1. It is good to see this method could cover different robot arms for the IK problem and leverage the symmetry of the representation.
2. It will be helpful if the vertices and the edges of $G$ and $\tilde{G}$ are defined clearly.
3. It will be appreciated if there are some analyses that why the GCN and GAT do not work at all.
4. It is more exciting to see how to apply this method to path planning problems instead of solving IK approximately.

---

### Official Review · Reviewer_a1Qk · 2023-06-17
**Distance-geometric representations and E(n) equivariance enable an IK solver for different manipulators - but what are its limitations?**

**Rating:** 8
**Confidence:** 4

**Review:**

This work proposes an E(n) equivariant GNN-based IK solver, using a distance-geometric representation to generate solutions for manipulators of varying structure and geometry.

Strengths:
- The presentation of the method is well structured, clearly motivating each design decision and why certain prior work is used.
- The symmetry of the mapping problem is leveraged by employing EGNNs [15], shown in an ablation study to be crucial for convergence.
- Fig. 1 gives a good intuition of the used representation and Fig. 3 nicely illustrates the considered breadth in terms of link geometry and DOF.

Weaknesses:
- I find the claim that the proposed approach "can be used with any manipulator" while other methods do not generalize to "robots that vary in link geometry and DOF" is not perfectly aligned with the evaluation setup. Would the method work on a previously unseen manipulator graph? Or on a known structure but with different distances?
- Somewhat in line with the previous comment, I believe that a paragraph on limitations or avenues for building upon this work is missing. For example, what are the method's training data requirements and runtimes compared to the baselines? Any failure cases that were observed? Hypotheses why a significantly higher error is observed for the Panda as compared to the others?
- There are some minor things in terms of presentation that, in my mind, would improve the clarity of the paper (see below).

I would recommend the authors to find space for a few lines on the limitations of the approach to also motivate readers to further investigate this interesting line work. I hope that the comments regarding the presentation will be helpful to further improve the clarity of the proposed work.

===
Comments on presentation:
- The definition of N and M is a bit unclear in Sec. II (mentioned only later in Alg. 1).
- In Fig. 1, the caption could mention color coding of distances, auxiliary points could use different markers and the base coordinate system seems inconsistent between Fig. 1 and 2.
- Fig. 2 could be improved to better indicate what happens at train and test time, respectively. Also, there seems to be an edge missing towards the eef joint.
- CVAEs should be defined at some point (or at least written out) to facilitate understanding by readers unfamiliar with generative models.
- In Tab. 1, the results on the Panda should be presented below one another. Finally, where are the results for baselines taken from? The respective paper? This is not perfectly clear to me by reading the caption alone.

---

### Decision · Program_Chairs · 2023-06-23

**Decision:**

Accept

**Comment:**

Congratulations! We encourage the authors to revise the paper based on the reviewer's feedback.
Your paper will be presented as both a short presentation and a poster. Detailed instructions about the presentation format and camera-ready submission will be sent to you soon.